# Impact of a 25% Salt Reduction on the Microbial Load, Texture, and Sensory Attributes of a Traditional Dry-Cured Sausage

**DOI:** 10.3390/foods9050554

**Published:** 2020-05-01

**Authors:** Miguel Elias, Marta Laranjo, Maria Eduarda Potes, Ana Cristina Agulheiro-Santos, Maria José Fernandes, Raquel Garcia, Maria João Fraqueza

**Affiliations:** 1MED-Mediterranean Institute for Agriculture, Environment and Development, IIFA-Instituto de Investigação e Formação Avançada, Universidade de Évora, Pólo da Mitra, Ap. 94, 7006-554 Évora, Portugal; elias@uevora.pt (M.E.); mlaranjo@uevora.pt (M.L.); mep@uevora.pt (M.E.P.); acsantos@uevora.pt (A.C.A.-S.); raquelg@uevora.pt (R.G.); 2Departamento de Fitotecnia, Escola de Ciências e Tecnologia, Universidade de Évora, Pólo da Mitra, Ap. 94, 7006-554 Évora, Portugal; 3Departamento de Medicina Veterinária, Escola de Ciências e Tecnologia, Universidade de Évora, Pólo da Mitra, Ap. 94, 7006-554 Évora, Portugal; 4CIISA-Centro de Investigação Interdisciplinar em Sanidade Animal, Faculdade de Medicina Veterinária, Universidade de Lisboa, Avenida da Universidade Técnica, 1300-477 Lisboa, Portugal; mjfernandes@fmv.ulisboa.pt

**Keywords:** meat sausages, *Catalão*, low salt, pig breed, quality, safety, fatty acids profile

## Abstract

*Catalão* is a Portuguese dry-cured traditional sausage, highly appreciated for its distinctive sensory properties. The aim of this study was to evaluate the impact of a 25% salt reduction on *Catalão* manufactured with either purebred Alentejano (Al) or crossbred Iberian × Duroc (IDr) pork meat, on its physicochemical and microbiological stability, texture parameters, and sensory attributes. No significant effect of salt reduction or genotype was observed for pH, a_W_, and microbiological parameters. PUFA content was significantly higher for Al *Catalão*, particularly due to the content in linoleic and linolenic fatty acids. IDr 3% NaCl samples had the highest mean n6/n3 PUFA ratio, and the highest mean values for the atherogenicity and thrombogenicity indices, showing that both genotype and salt content influence these nutritional indices. Texture profile of *Catalão* was significantly influenced by salt content and genotype. Al samples were less adhesive, cohesive, and easier to chew. Low-salt *Catalão* was harder, more adhesive, and less cohesive, with lower resilience and higher chewiness values. Regarding sensory attributes, salt content influenced the product aroma, with reduced-salt sausages being evaluated as significantly less aromatic. Overall, a 25% salt reduction did not have a significant impact on the quality, stability, and sensory evaluation of *Catalão*.

## 1. Introduction

Traditional cured meat products are appreciated worldwide because of their sensory characteristics. These sensory characteristics are related to meat selection, recipe, and processing of sausages among other factors [1]. In the meat selection for high quality sausages, meat from indigenous pig breeds raised under extensive or semi-extensive conditions is chosen for their more suitable color, their interspersed and infiltrated fat between the muscular fiber (intramuscular fat, marbling), and their firmer texture [2]. Moreover, autochthonous pig breeds are often fed with local products as is the case with the acorn-finished pigs, which provides unique characteristics to the meat, and particularly to the fat [2,3].

Traditional dry-cured meat sausages have been considered delicatessen products, but have lately been regarded as potentially unhealthy foods for several consumers, due to nutritional characters, associated to their high fat and salt contents [4,5,6]. High salt content in foods has been referred as a health factor risk for human inducing high blood pressure and cardiovascular diseases [7,8]. The reduction of salt on cured meat products is a task requested to the meat industry by inputs given by human health organizations and other related institutions (World Health Organization—WHO, Pan American Health Organization—PAHO, European Centre for Disease Prevention and Control-ECDC). The World Health Organization (WHO) has supported the implementation of national salt reduction strategies over the last two decades [9]. For several food products, the goal should be a reduction of at least 25% in salt content, considering their usual average. However, the perception of salt by consumers could have an impact on their preferences, because food products perceived as unsalted or very low in salt could be depreciated. Meat products are complex products, in which each ingredient plays a specific role, making their reformulation a challenging task [1]. The impact of salt reduction in the curing process of sausages depends on each product, with its indigenous fermentative microbiota, and could influence the microbial ecosystem with deviation of sensory attributes, including texture, and on the edge resulting in spoilage or unsafe products.

The production of low-salt, dry-cured meat sausages must assure not only safe and stable products, but also that consumers’ appreciation will be not affected. Previous studies on *Catalão* sausages [10] had already suggested that salt content could be reduced to 3% NaCl in the final product, with the consequent health-related advantages. Furthermore, those authors also reported that salt content had the greatest impact on the studied parameters, compared to genotype [10].

The aim of the current work was to evaluate the impact of a 25% salt reduction on *Catalão,* a dry-cured meat traditional sausage produced with either purebred or crossbred Iberian pork meat, on its stability (physicochemical and microbiological) and texture, as well as on its sensory attributes.

## 2. Materials and Methods

### 2.1. Processing and Sampling of a Cured Meat Sausage

*Catalão,* a cured traditional meat sausage was manufactured in a local factory of the Alentejo region (Portugal) with two different meat raw materials, either Alentejano purebred (Al) or crossbred Iberian × Duroc (IDr) pork meat.

Meat was manually cut into large pieces, mechanically minced (5.0 × 5.0 mm), and mixed with 5% minced backfat (5.0 × 5.0 mm), white wine (3.50%), sodium chloride (NaCl), black pepper (*Piper nigrum* L.) (0.15%), white pepper (0.15%), cumin (*Cuminum cyminum* L.) (0.10%), disodium diphosphate (0.04%), pentasodium triphosphate (0.04%), NaNO_3_ (0.003%), and KNO_2_ (0.003%). Nitrates and nitrites were added in the form of the commercial additive NITROS 5/5 (Formulab, Portugal). Regarding salt content, *Catalão* sausages were manufactured with the commercial formulation usually (4% salt in the final product) and with a 25% salt reduction (3% in the final product). Therefore, four different groups of sausages were produced: Al 3% NaCl, Al 4% NaCl, IDr 3% NaCl, and IDr 4% NaCl.

After two days under refrigeration at 5 °C and 90% relative humidity, for ripening purposes, the meat batters were stuffed into cleaned and desalted natural pork casings, with 36–38 mm. *Catalão* sausages weight around 150 g (final product) with a horseshoe shape.

The general curing procedure of meat sausages occurred in two steps: smoking and drying. The sausages were indirectly exposed to smoke generated by burning oak (*Quercus ilex* L.) wood, for two days, approximately 7 h/day. The drying process took place in a controlled environment chamber (7 °C and 80–85% relative humidity) until a 35% weight loss was reached, which took about 18 days. Three independent production batches of *Catalão* were manufactured for each group, with duplicate samples (*Catalão* sausages) being collected after 35% weight loss (final product). Sausages were immediately processed for physicochemical, microbiological, and sensory analyses.

### 2.2. Microbiological Analyses

Sausages were prepared for microbial analysis according to ISO 6887-2 [11]. Microbiological analyses were performed following established ISO procedures or previously described methods: *Salmonella* spp. detection (ISO 6579) [12]; *Listeria monocytogenes* (ISO 11290-2) [13]; mesophiles (4833-1) [14]; lactic acid bacteria (LAB) (ISO 15214) [15]; staphylococci [16]; enterococci as described by Talon et al. [17]; *Escherichia coli* (ISO 16649-2) [18]; and molds and yeasts (ISO 21527-2) [19]. All microbiological analyses were performed in triplicate and the results were expressed in log cfu/g.

### 2.3. Physicochemical Analyses

#### 2.3.1. Determination of pH and a_W_

pH values were measured in triplicate in each homogenized sample with a pH meter (HI 9025; electrode FC 230B) equipped with a pH electrode (FC 230B, Hanna Instruments, USA) according to the procedures described in ISO 2917 [20]. Water activity (a_W_) was determined using a Rotronic Hygrometer station (Rotronic Hygroskop DT) previously calibrated at 20 ± 1 °C with EA00-SCS, EA50-SCS and EA80-SCS Humidity Standards (Rotronic, Ettlingen, Germany).

#### 2.3.2. Determination of Total Chloride Content

Total chloride content was quantified according to the Volhard method (ISO 1841-1) [21] and expressed as sodium chloride as a percentage by mass.

#### 2.3.3. Analysis of Fatty Acids Profile

Each sausage was cut into pieces and grounded in a mechanical mill, lyophilized, placed in a glass flask, and stored at 4 °C until use. The extraction of fatty acids from *Catalão* sausages was performed using a Dionex 100 accelerated solvent extractor (ASE) (Dionex Corporation, USA) using the following extraction procedure: aliquots of approximately 300 mg of sausages was combined with 6 g diatomaceous earth (Dionex Corporation, USA), and the mixture was transferred to a 34 mL stainless steel extraction cell fitted with two cellulose filters. Further, the extraction procedure carried out with a mixture of chloroform/methanol (60:40 (*v*/*v*) (Merck, Germany) containing 100 mg·L^−1^ BHT (3,5-di-tert-butyl-4-hydroxytoluene) (Merck, Germany) as an antioxidant at 100 °C and 12.4 MPa. Two static extraction cycles were carried out for 5 min each. The solvent was removed under vacuum using a Vacobox B-177 (Buchi, Switzerland) equipped with a vacuum controller B-720, a rotavapor R-114 (Buchi, Switzerland) attached to a water bath B-480 afforded a crude residue. Next, crude residue was dissolved with 1 mL of chloroform and an aliquot of 100 µL was taken into a glass tube and the solvent was removed under a stream of nitrogen. The residue obtained was saponified in methanolic NaOH solution (0.5 N) at 70 °C for 15 min. Fatty acids were derivatized with boron-trifluoride-methanol solution (10 g BF3 L^1^CH_3_OH) (Merck, Germany) in order to give fatty acids methyl esters (FAMEs) as described by Morrison and Smith [22]. FAMEs were analyzed by gas chromatography in a Hewlett Packard HP 6890 Series GC System (HP, USA) equipped with a split-splitless injector, an auto-sampler, a flame-ionization detector (FID), an Omegawax 320 fused silica capillary column (30 m, 0.32 mm i.d., 0.25 mm film thickness) (Supelco, USA), and HPChem software (2002). The chromatographic conditions were according to Laranjo et al. [23]. FAMEs were identified by comparison of their retention times with known standards (37-component FAME mix, Supelco 47885-U) chromatographed in identical gas chromatography conditions and by the determination of Kovats indices (data not shown). Relative fatty acid composition was quantified for each sample and presented as percentage weight for fatty acid composition.

#### 2.3.4. Texture Profile Analysis

Texture Profile Analysis (TPA) was performed at room temperature (20 ± 1 °C) on five replicates per sample using a Stable Micro System TA-Hdi (Stable Micro Systems, Godalming, England) as described previously [23,24,25]. Using the obtained force time curves, the resulting parameters were evaluated/calculated: hardness, adhesiveness, springiness, cohesiveness, resilience, and chewiness [26].

### 2.4. Sensory Analysis

Sensory analysis was performed at the University of Évora, Portugal, in a Sensory Test Room prepared according to the environment requirements described in ISO 8589 [23]. Selected assessors were chosen, trained, and monitored following the procedures described in ISO 8586 [27] for this specific meat product. Sample preparation and analysis were performed as described by Fraqueza et al. [28]. Each assessor evaluated a maximum of six samples per session. A Quantitative Descriptive Analysis (QDA) with 12 attributes was used [29]. All attributes ranged from 0 (“not perceived”) to 100 (“maximum perception”). For the attributes “hardness” and “salt perception”, 50 was considered as the optimum value of the QDA scale.

### 2.5. Data Analysis

Data analysis was performed using Statistica version 12 software (StatSoft, Inc., USA). A total of 24 *Catalão* sausages (4 groups × 3 batches × 2 samples) were analyzed. Data were presented as means ± standard error deviation. Analyses of variance (ANOVA) were performed for the factors genotype and salt content.

## 3. Results and Discussion

No significant differences were observed between batches, proving that the authors effectively analyzed three independent production batches. Consequently, the factor batch was not further considered for statistical analysis, and all replicates were analyzed together. Moreover, no significant interaction was observed between the two factors, genotype and salt content, and thus, one-way ANOVAs were performed for each factor separately.

### 3.1. Effect of Pig Genotype and Salt Reduction on Physicochemical and Microbial Parameters of Catalão

*Catalão* cured sausages had a pH range from 5.46 to 5.60 (Table 1), independently of pig genotype or salt content. These products presented a low a_W_, varying between 0.848 and 0.854. These values were not influenced by the breed of raw meat in use and neither by salt content resulting from the process control of drying that achieved a loss weight of approximately 35%. The content of total chlorides was significantly different in sausages formulated for salt reduction. These products in the market are usually formulated with a higher salt content of approximately 4–5%. The salt reduction preconized in this study can be achieved with a stabilized product being observed that *E. coli*, *Salmonella* spp. and *Listeria monocytogenes* were not present. The technological microbiota achieved high counts independently of the raw meat breed used or salt content formulated. LAB and enterococci present on the final product *Catalão* ranged between 8.1 and 8.6 log cfu/g. This microbiota is involved in meat fermentation present on traditional sausages and responsible by the production of lactic acid and other metabolites (bacteriocins) implicated in the protection of the meat product [30]. Furthermore, staphylococci and yeasts achieved higher counts (5.9–7.0 and 6.5–7.7 log cfu/g, respectively) in the final product, probably because they are participating in the development of sausage sensory characteristics as referred by some authors [31,32,33].

### 3.2. Effect of Pig Genotype and Salt Reduction on the Fatty Acids Profile of Catalão

The composition of fatty acids methyl esters of *Catalão* is summarized in Table 2.

As observed in Table 2, the major fatty acids present in *Catalão* sausages were oleic (C18:1), palmitic (C16:0), stearic (C18:0), and linoleic (C18:2) acid (Table 2), in decreasing concentration levels, which agrees with the results of other authors [23,34]. Oleic acid is the most abundant component of the lipid fraction [34,35] and it does not show any significant changes with genotype (Table 1). However, the low values obtained for sausages manufactured with Alentejano pork meat are probably due to feeding reasons [3,34]. *Catalão* sausages manufactured with Alentejano pig meat had significantly higher contents of PUFA linoleic (C18:2) and linolenic (C18:3) acids (*p* < 0.05), essential fatty acids for human health, that must be provided by the diet, because they cannot be synthesized by humans [36].

Salt content did not have any influence on the fatty acid profile of *Catalão* sausages (Table 2).

SFA contents ranged between 30.97% and 36.59% and MUFA concentrations were between 45.11% and 47.76% (Table 2), with no significant differences between genotypes or salt concentrations. The content in PUFA was significantly higher (*p* < 0.001) for *Catalão* sausages made with Alentejano (Al) raw meat (8.66–8.97%), particularly due to the content of linoleic (C18:2) and linolenic (C18:3) fatty acids (Figure 1).

Based on the results obtained from the FAMEs profile of *Catalão*, the n6/n3 PUFA ratio and some nutritional indices, namely the atherogenicity (AI) and the thrombogenicity (TI) indices, were also calculated according to Larqué and co-workers [37]. According to the literature, it is assumed that these two health-related lipid indices are the strongest markers in predicting some diseases, like cardiovascular diseases. Results are listed in Table 3.

Concerning n6/n3 PUFA ratios, high values were obtained (14.94–19.78), since the recommended n6/n3 PUFA ratio should not exceed a value of 4 [38]. Scientific evidence suggests that a very high n6/n3 PUFA ratio promotes the pathogenesis of many diseases, including cardiovascular diseases, cancer, and inflammatory and autoimmune diseases [38].

The Atherogenicity Index (AI) expresses the relationship between the sum of the main SFA (pro-atherogenic) and that of the main MUFA and PUFA (anti-atherogenic) [39]. The AI varies from 0.70 to 0.80 for the *Catalão* samples studied in this work, these values being very similar to other AI reported for fermented sausages made of pork meat from various breeds [40].

The Thrombogenicity Index (TI) can be defined as the ratio between the pro-thrombogenic SFA and the anti-thrombogenic MUFA and PUFA [39]. Regarding TI, the calculated values were lower than others reported in the literature [38]. Overall, IDr 3% NaCl samples showed the highest mean values for all three indices. The genotype seems to have an influence on TI (*p* < 0.001). On the other hand, AI and TI show higher values for lower salt concentrations (*p* < 0.05).

The values obtained for both AI and TI are generally low, when compared to the literature [39,41,42]. In fact, olive oil, which is rich in MUFA, mainly oleic acid (C18:1), has values of 0.14 and 0.32 for AI and TI, respectively [39]. In the same study, values of 0.58–0.69 and 1.35–1.66, respectively for AI and TI, were reported for pork lean and fatty meat and meat sausages [39].

Despite of the significant differences observed for Atherogenicity and Thrombogenicity Indices associated to an increased salt content, the relative abundances of SFA and MUFA do not show significant differences with salt content (*p* > 0.05). Therefore, no major differences were found in the fatty acids profile regarding salt content, that may justify the differences depicted in these indices.

### 3.3. Effect of Pig Genotype and Salt Reduction on Texture and Sensory Evaluation of Catalão

Texture profile of *Catalão* was significantly influenced by pig genotype, particularly the following parameters: adhesiveness (*p* < 0.001), cohesiveness (*p* < 0.01), springiness and chewiness (*p* < 0.05) (Table 4). Cured meat products made with Alentejano pig had lower adhesiveness, cohesiveness, springiness and chewiness than those made with meat from commercial Iberian × Duroc crossbreed pigs. Salt concentration also had a significant effect on the texture of *Catalão* (*p* < 0.05) (Table 4). When salt concentration was reduced to 3%, the cured meat product was harder, more adhesive, and less cohesive, giving a lower resilience and a higher chewiness. In fact, other authors reported the influence of pig genotype on cured sausage texture with Iberian × Duroc crossbreed pig sausage with higher hardness and adhesiveness due to the resistance of their muscular fibers and distribution of the fat [23,43]. Salt influences the solubility of meat proteins. A higher salt content leads to an increase in the soluble protein fraction, contributing to an improved binding of the meat batters, and thus to significantly higher cohesiveness values for sausages with a 4% NaCl content.

Regarding sensory analysis, the results of different attributes evaluated by the panelists are shown in Table 5.

The main differences detected by the sensory panel in *Catalão* sausages (Figure 2 and Table 5) were related to the marbled aspect of this cured product, with higher values for the sausages manufactured with the commercial Iberian × Duroc crossbreed. This contradicts other authors stating that cured loins made with crossbreeds Iberian × Duroc pigs presented less marbling and lower intramuscular fat (IMF) content [43]. Regarding fibrousness, significantly higher values were reported for Alentejano sausages.

Salt content influenced the product aroma. Products with lower salt content (3%) presented significantly lower aroma intensity (*p* < 0.05), already reported in previous studies [10,44,45,46]. Corral et al. [46] also referred that cured products with a reduction of 16% salt presented a reduction of sulphur and acid and increased aldehyde compounds.

Besides a lower aroma intensity, no other significant differences were perceived by assessors regarding reduced-salt *Catalão* sausages. This may be considered an advantage, because it means that the product was not unsalted or tasteless. All in all, reduced-salt *Catalão* sausages are stable without significant changes in the analyzed microbial groups, without a significant damage in texture or fatty acids profile. Therefore, a 25% salt reduction, with reference to the salt content of the original *Catalão* sausages available in the market, can be beneficial, provided that manufacturing procedures, leading to a weight loss of 35% in the final product, are observed.

## 4. Conclusions

The effect of salt reduction and genotype on the quality and safety of dry-cured *Catalão* sausages was studied. Generally, a 25% salt reduction did not have a significant impact on the stability, safety, and sensory appreciation of *Catalão* sausages, which thereby retained their quality.

Furthermore, no significant effect of salt reduction or genotype was observed for pH, a_W_, and microbiological parameters. Regarding genotype, the content in PUFA was significantly higher for Alentejano *Catalão* sausages, particularly due to the content in linoleic and linolenic fatty acids. Both genotype and salt content had influence in the n6/n3 PUFA ratio, and the atherogenicity and thrombogenicity indices, with IDr 3% NaCl samples showing the highest mean values for all three indices. Texture profile of *Catalão* was significantly influenced by salt content, with Alentejano samples being less adhesive, cohesive, and easier to chew. Low-salt *Catalão* sausages were harder, more adhesive, and less cohesive, with lower resilience and higher chewiness values. The texture profile of *Catalão* was also influenced by genotype, with significantly lower adhesiveness, cohesiveness, springiness and chewiness values for the Alentejano sausages. Regarding sensory attributes, salt content influenced aroma intensity, with reduced-salt sausages being evaluated as significantly less aromatic, which had been reported before for similar products.

Based on the presented results, and following the WHO recommendations, sausage manufacturers may be advised to safely reduce salt content, without compromising their safety and sensory attributes.

## Figures and Tables

**Figure 1 foods-09-00554-f001:**
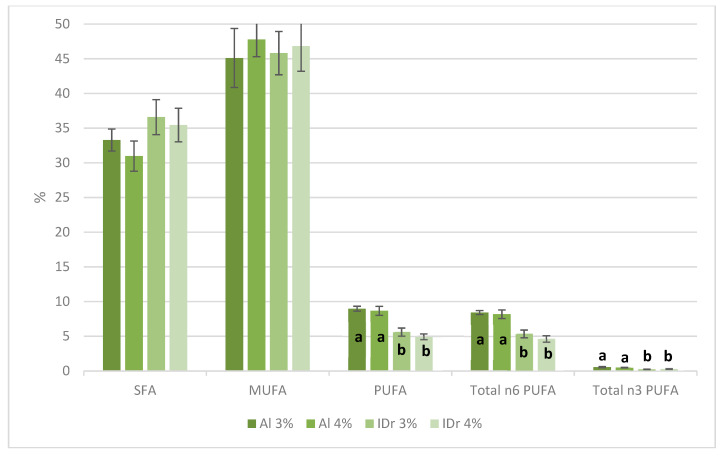
*Catalão* methyl ester fatty acid-derivatives (expressed in %), with PUFA content showing significant differences regarding genotype.

**Figure 2 foods-09-00554-f002:**
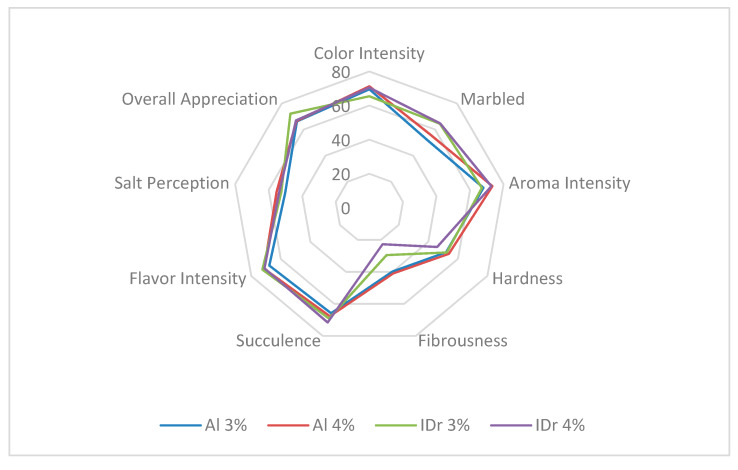
Graphic representation of the *Catalão* sensory evaluation by quantitative descriptive analysis according to genotype and salt content.

**Table 1 foods-09-00554-t001:** Influence of pig genotype and salt content (% NaCl) in physicochemical parameters and microbial counts of *Catalão*.

Parameters	Al		IDr		*p* Value	
	3% NaCl	4% NaCl	3% NaCl	4% NaCl	G	S
pH	5.46 ± 0.06	5.46 ± 0.06	5.54 ± 0.04	5.60 ± 0.06	0.067	0.586
a_W_	0.854 ± 0.010	0.862 ± 0.009	0.877 ± 0.012	0.848 ± 0.012	0.669	0.330
Chlorides	3.37 ^B^ ± 0.17	3.99 ^A^ ± 0.17	3.26 ^B^ ± 0.06	4.11 ^A^ ± 0.25	0.985	0.000 ***
Mesophiles	8.4 ± 0.1	8.5 ± 0.1	8.4 ± 0.0	8.4 ± 0.1	0.562	0.590
Staphylococci	6.0 ± 0.1	6.1 ± 0.4	5.9 ± 0.1	7.0 ± 0.7	0.432	0.177
LAB	8.5 ± 0.1	8.6 ± 0.1	8.5 ± 0.0	8.4 ± 0.0	0.185	0.809
Enterococci	8.3 ± 0.1	8.1 ± 0.3	8.3 ± 0.0	8.3 ± 0.0	0.430	0.497
Yeasts	6.5 ^b^ ± 0.3	6.8 ^b^ ± 0.2	7.7 ^a^ ± 0.1	7.4 ^a^ ± 0.1	0.003 **	0.967

Data are represented as mean ± standard error of the mean (SEM). Microbial counts are expressed as log cfu/g. Al—purebred Alentejano pig; IDr—commercial Iberian × Duroc crossbred pig. G—genotype; S—salt content; LAB—lactic acid bacteria. Significance: ** *p* < 0.01, *** *p* < 0.001. In each row: different small letters represent significantly different means for genotype factor (Tukey’s HSD test); different capital letters represent significantly different means for salt content (Tukey’s HSD test).

**Table 2 foods-09-00554-t002:** Influence of pig genotype in methyl ester fatty acid-derivatives profile of *Catalão*.

Fatty Acids	Al		IDr		*p* Value
	3% NaCl	4% NaCl	3% NaCl	4% NaCl	
Lauric-C12:0	0.23 ± 0.12	0.13 ± 0.04	0.16 ± 0.04	0.35 ± 0.13	0.431
Myristic-C14:0	2.92 ^a^ ± 0.44	3.00 ^a^ ± 0.07	1.77 ^b^ ± 0.45	0.53 ^b^ ± 0.01	0.000 ***
Palmitic-C16:0	20.59 ± 1.12	19.11 ± 1.36	22.78 ± 1.70	22.63 ± 1.48	0.081
Palmitoleic-C16:1	2.76 ± 0.27	2.63 ± 0.16	2.57 ± 0.18	2.73 ± 0.22	0.845
Margaric-C17:0	0.30 ± 0.02	0.22 ± 0.05	0.36 ± 0.10	0.34 ± 0.05	0.182
Margaroleic-C17:1	0.27 ± 0.02	0.27 ± 0.01	0.42 ± 0.14	0.51 ± 0.11	0.063
Stearic-C18:0	9.25 ^b^ ± 0.17	8.51 ^b^ ± 0.83	11.52 ^a^ ± 0.84	11.60 ^a^ ± 0.93	0.007 **
Oleic-C18:1	40.61 ± 3.79	43.38 ± 2.27	41.91 ± 3.03	42.51 ± 3.27	0.948
Linoleic-C18:2	8.42 ^a^ ± 0.29	8.17 ^a^ ± 0.62	5.34 ^b^ ± 0.55	4.63 ^b^ ± 0.45	0.000 ***
Linolenic-C18:3	0.56 ^a^ ± 0.09	0.49 ^a^ ± 0.04	0.27 ^b^ ± 0.04	0.31 ^b^ ± 0.05	0.004 **
Gadoleic-C20:1	0.87 ± 0.39	1.34 ± 0.07	0.85 ± 0.09	0.67 ± 0.34	0.226
Heneicosanoic-C21:1	0.60 ± 0.50	0.14 ± 0.04	0.06 ± 0.03	0.39 ± 0.39	0.656

Data are represented as mean ± standard error of the mean (SEM). Fatty acids are expressed in percentage (%). Al—purebred Alentejano pig; IDr—commercial Iberian × Duroc crossbred pig. Significance: ** *p* < 0.01, *** *p* < 0.001. In each row: different small letters represent significantly different means for genotype factor (Tukey’s HSD test).

**Table 3 foods-09-00554-t003:** Nutritional indices for *Catalão* samples according to genotype and salt content.

Indices	Al		IDr		*p* Value	
	3% NaCl	4% NaCl	3% NaCl	4% NaCl	G	S
n6/n3 PUFA ratio	15.86 ± 2.19	16.85 ± 0.40	20.25 ± 2.48	16.21 ± 4.51	0.522	0.601
AI	0.77 ^AB^ ± 0.01	0.70 ^B^ ± 0.01	0.81 ^A^ ± 0.03	0.70 ^B^ ± 0.01	0.339	0.002 **
TI	0.71 ^bA^ ± 0.02	0.62 ^bB^ ± 0.02	0.79 ^aA^ ± 0.01	0.76 ^aB^ ± 0.01	0.000 ***	0.007 **

Values are represented as mean ± standard error of the mean (SEM). Significance: ** *p* < 0.01, *** *p* < 0.001. In each row: different small letters represent significantly different means for genotype factor (Tukey’s HSD test); different capital letters represent significantly different means for salt content (Tukey’s HSD test). Indices are expressed in percentage (%). Atherogenicity Index: AI = [(4 × C14:0 + C16:0) + C18:0]/[∑MUFA + ∑PUFA-n6 + ∑PUFA-n3]; Thrombogenicity Index: TI = (C14:0 + C16:0 + C18:0)/[(0.5 × C18:1 + 0.5 × ∑MUFA + 0.5 n6PUFA + 3 × n3PUFA + (n3PUFA/n6PUFA)] [37].

**Table 4 foods-09-00554-t004:** Influence of pig genotype and salt content (% NaCl) in the texture profile analysis of *Catalão*.

Texture Parameters	Al		IDr		*p* Value	
	3% NaCl	4% NaCl	3% NaCl	4% NaCl	G	S
Hardness (N)	53.995 ^A^ ± 2.743	35.664 ^B^ ± 2.777	52.671 ^A^ ± 2.739	38.501 ^B^ ± 2.683	0.783	0.000 ***
Adhesiveness (N*s)	−2.513 ^bA^ ± 0.399	−1.557 ^bB^ ± 0.433	−4.642 ^aA^ ± 0.490	−3.306 ^aB^ ± 0.337	0.000 ***	0.008 **
Cohesiveness	0.537 ^bB^ ± 0.012	0.601 ^bA^ ± 0.018	0.578 ^aB^ ± 0.013	0.651 ^aA^ ± 0.008	0.001 **	0.000 ***
Springiness (mm)	0.808 ^b^ ± 0.020	0.862 ^b^ ± 0.026	0.885 ^a^ ± 0.021	0.876 ^a^ ± 0.013	0.032 *	0.292
Resilience	0.141 ^B^ ± 0.006	0.181 ^A^ ± 0.017	0.138 ^B^ ± 0.005	0.172 ^A^ ± 0.004	0.533	0.000 ***
Chewiness (N*mm)	23.553 ^bA^ ± 1.542	18.261 ^bB^ ± 1.491	26.817 ^aA^ ± 1.407	22.080 ^aB^ ± 1.782	0.027 *	0.002 **

Data are represented as mean ± standard error of the mean (SEM). Al—purebred Alentejano pig; IDr—commercial Iberian × Duroc crossbred pig. G—genotype; S—salt content. Significance: * *p* < 0.05, ** *p* < 0.01, *** *p* < 0.001. In each row: different small letters represent significantly different means for genotype factor (Tukey’s HSD test); different capital letters represent significantly different means for salt content (Tukey’s HSD test).

**Table 5 foods-09-00554-t005:** Influence of pig genotype and salt content in the sensory evaluation of *Catalão*.

Sensory Attributes	Al		IDr		*p* Value	
	3% NaCl	4% NaCl	3% NaCl	4% NaCl	G	S
Color intensity	70 ± 4	71 ± 3	65 ± 2	71 ± 2	0.395	0.237
Off colors	1 ± 0	0 ± 0	1 ± 0	0 ± 0	0.682	0.066
Marbled	52 ^b^ ± 4	56 ^b^ ± 4	64 ^a^ ± 4	65 ^a^ ± 3	0.009 **	0.623
Aroma intensity	68 ^B^ ± 3	73 ^A^ ± 3	67 ^B^ ± 3	73 ^A^ ± 2	0.772	0.037 *
Off aromas	1 ± 0	2 ± 1	1 ± 0	0 ± 0	0.199	0.305
Hardness	52 ± 3	54 ± 3	52 ± 2	46 ± 2	0.126	0.350
Fibrousness	40 ^a^ ± 5	41 ^a^ ± 6	30 ^b^ ± 4	23 ^b^ ± 3	0.003 **	0.561
Succulence	66 ± 3	68 ± 3	69 ± 3	72 ± 2	0.176	0.424
Flavor intensity	68 ± 4	71 ± 3	72 ± 2	71 ± 3	0.475	0.722
Off flavors	3 ± 1	1 ± 1	2 ± 2	1 ± 1	0.861	0.259
Salt perception	50 ± 2	55 ± 2	52 ± 1	54 ± 3	0.875	0.167
Overall appreciation	66 ± 3	66 ± 4	72 ± 3	67 ± 4	0.332	0.467

Data are represented as mean ± standard error of the mean (SEM). Al—purebred Alentejano pig breed; IDr—commercial Iberian × Duroc crossbred pig. Significance: * *p* < 0.05, ** *p* < 0.01. In each row: different small letters represent significantly different means for genotype factor (Tukey’s HSD test); different capital letters represent significantly different means for salt content (Tukey’s HSD test).

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
