# Peer review of "Impact of a 25% Salt Reduction on the Microbial Load, Texture, and Sensory Attributes of a Traditional Dry-Cured Sausage"

_foods, 2020, doi:10.3390/foods9050554_

Round 1
Reviewer 1 Report
There are many parameters that influence the quality of Catalão (Portuguese dry-cured traditional sausage): special microbiota of the product, salt content, physico-chemical and sensory attributes etc. The research article introduces to examining the impact of a 25 % salt reduction for parameters of quality.
In my opinion the article is too short. The introduction and the conclusion are too short. The conclusion section is too short, it is worth comparing with the international results. The summary is too general, more test results should be included. In my experience, you might want to enter keywords that are not in the title for keywords. The part of introduction and practical applications are right. In abstarct, the authors write: „Regarding sensory attributes, salt content influenced the product aroma, with reduced-salt sausages being evaluated as less aromatic.” Is this significantly proven by the results? (ANOVA, Tukey-HSD).
The structure of the article is adequate. The materials and methods part is sufficiently detailed. Statistical methods and software are well presented. Unfortunately, however, I did not find the results of PCA in the manuscript.
This section is part of the results section: „No significant differences were observed between batches. Therefore, the factor batch was not considered, and all replicates were analysed together. Since interactions were not significant, one-way ANOVAs were performed for each factor separately. Principal Components Analyses (PCA) were also carried out.”
In the methodology, the analytical part is accurate, but it is not accurate enough to describe the sensory evaluations. This research should include the following information: name of the sensory laboratory, duration of sessions, reference to the International Standard (ISO 13299, ISO 8589, valid document: ISO 8586:2012 Sensory analysis -- General guidelines for the selection, training and monitoring of selected assessors and expert sensory assessors, and quality control procedures), definition of replicates (same/different day(s), same/different batch(es)), type of assessors (selected assessors or experts); attributes definition, their protocol of evaluation, and the reference substances, etc. What was the specific training for teas descriptors? What methodologies were used for the panel/panelist performance?
The sensory subchapter needs to be expanded to include the selection steps and performance of the reviewers. A good sensory panel should provide accurate, discriminate and accurate results. Ideal group performance is achieved when each group member distinguishes between products (high product variabilty), achieves the same values ​​several times ("low within sessions"). However, he agrees with the other panel members on the sensory characteristics within a given tolerance (small difference between panelist’). By monitoring and monitoring performance, panel members and panels can be distinguished, their results are permanent, repeatable. Performance can be measured with performance indicators. The quality of the sensory data is determined by a qualified and expert sensory panel, and its members, and their performance must be constantly monitored. This is best done with software support, for example: PanelCheck V1.4.0., Sensominer (R-project), etc.
How did the Adhesiveness property get negative (Table 2)?
I found some inaccurate phrases or fonts that I suggest to improve:
- In tables, values for a single parameter should be given to the same number of decimal places everywhere.
- 2.L.47.: World Health Organization instead of World Health Organisation
- 2.L.49.: World Health Organization instead of World Health Organisation
- 2.L.71.: 3.50% instead of (3,5%)
Author Response
Mr. Emil-Gabriel Nutu
Assistant Editor
Foods
Lisbon, April 8, 2020
Dear Mr. Emil-Gabriel Nutu,
Thank you for considering our manuscript, Foods-772743, entitled “Impact of a 25% salt reduction on the microbial load, texture and sensory attributes of a traditional dry-cured sausage”, by Miguel Elias, Marta Laranjo, Maria Eduarda Potes, Ana Cristina Agulheiro-Santos, Maria José Fernandes, Raquel Garcia, and Maria João Fraqueza.
First, regarding the Plagiarism Report of the manuscript, the higher matches are restricted to the Materials and Methods section, and all original papers are properly cited. Still, we have tried to rephrase some sentences and parts of the article in order to decrease the repetition rate, as requested.
We have revised the manuscript according to the comments made by the editor and the two reviewers. Their suggestions have been addressed and modifications have been included in the revised version with the track changes function of the word processor. We hope that the changes now included in the manuscript meet your expectations. If any additional clarification or revision is needed, please let us know.
Editor:
1- Authors included some information about the “batches” and samples, however it is so confusing (lines 86-88). In all researchers authors must include samples (each sample that authors analyzed), batches (the manufacture/formulation/breeds,… are different), and replicates (number of times the experiment was replicated). As I understand, in this study there are 4 batches (2 different genotypes (Alentejo and DurocxAlentejo) and 2 salt contents (3 and 4%), in total 4 batches). Then, authors said: “Three independent batches of…”. The authors must say replicates, the manufacture process replicates?. The triplicate is not the number of samples, is the number of manufacture process. That is, the number of times the experiment was replicated, including all manufacture steps, batches, and samples within each batch. Authors said in the same lines “…duplicate samples”. Authors only analysed 2 samples from each batch and manufacture replicate?. At least, in this type of researchers, 15 samples must be analysed. The number of samples is very important to determine the validity of the study and the representativeness of the data, and therefore, determine if the manuscript has adequate design and if it is capable of being sent for review or not. Moreover, in this case, I think that a sentence summarized all experimental design must be included in the “2.5. Data analysis” section. My suggestion is that the authors include something like that: “A total of XXX samples were analysed [(## sausages x 4 batches x 3 (triplicate manufacture)]”, and delete the lines 86-88.
-We tried to clarify the information on batches and samples (lines 95-96 and 237-238).
2- The authors said in different lines “supplementary material” (Table S1 and Table S2). Why supplementary? Please, include all the information in the manuscript. Foods have no word limits, thus all relevant data must be included in the manuscript. If the data are no relevant for the manuscript they must be avoided. In fatty acids section, please, include all fatty acid detected in the samples. Also must include the SFA, MUFA, PUFA, n-3 and n-6 summatories. Additionally, I encourage authors calculate the nutritional indices as atherogenic index, thrombogenic index, hypercholesterolemic/hypochoresterolemic FA ratio and the n-6/n-3 ratio.
-Tables S1 and S2 have now been included in the manuscript. The requested summatories and nutritional indices have been calculated and included in manuscript in Table 2, Figure 1 and Table 3.
3- Result and discussion section. I think this is the biggest drawback that the manuscript presents before being sent for review. The authors only describe the data, and at best compare it to a couple of authors. However, in all sections the discussion part is missing. Therefore, from the results, the authors should conveniently discuss them, not just compare them with other authors. This part requires a lot of work, since it is very poor. Several researchers were published with these topics (both, effect of genotype in meat quality and development/reformulation of meat products with salt reduction).
-Thank you for your suggestion, which we followed to improve the Results & Discussion section of the manuscript. We hope that the introduced changes meet your expectations.
4- Finally, the references, both, in text and reference list did not follow Journal guidelines. The authors must read carefully authors guidelines.
-Thank you for the remark. This was a mistake. All citations and references are not formatted according to the journal guidelines.
Reviewer #1:
In my opinion the article is too short. The introduction and the conclusion are too short. The conclusion section is too short, it is worth comparing with the international results. The summary is too general, more test results should be included. In my experience, you might want to enter keywords that are not in the title for keywords. The part of introduction and practical applications are right. In abstarct, the authors write: „Regarding sensory attributes, salt content influenced the product aroma, with reduced-salt sausages being evaluated as less aromatic.” Is this significantly proven by the results? (ANOVA, Tukey-HSD).
-The introduction and conclusions have been expanded. We have revised the keywords included in the manuscript. Yes, low-salt Catalão sausages are significantly less aromatic (please see lines 422-423 and Table 5).
The structure of the article is adequate. The materials and methods part is sufficiently detailed. Statistical methods and software are well presented. Unfortunately, however, I did not find the results of PCA in the manuscript.
-Thank you for the positive comments. No PCA was performed. I am truly sorry, but this was a mistake and the sentence on PCA has been removed from the manuscript.
This section is part of the results section: „No significant differences were observed between batches. Therefore, the factor batch was not considered, and all replicates were analysed together. Since interactions were not significant, one-way ANOVAs were performed for each factor separately. Principal Components Analyses (PCA) were also carried out.”
-The abovementioned sentences have been moved to the Results and Discussion section.
In the methodology, the analytical part is accurate, but it is not accurate enough to describe the sensory evaluations. This research should include the following information: name of the sensory laboratory, duration of sessions, reference to the International Standard (ISO 13299, ISO 8589, valid document: ISO 8586:2012 Sensory analysis -- General guidelines for the selection, training and monitoring of selected assessors and expert sensory assessors, and quality control procedures), definition of replicates(same/different day(s), same/different batch(es)), type of assessors (selected assessors or experts); attributes definition, their protocol of evaluation, and the reference substances, etc. What was the specific training for teas descriptors? What methodologies were used for the panel/panelist performance?
-The methodology regarding sensory evaluations has been described in more detail in the revised version of the manuscript. Specifically, the referred International Standards (ISO) have been added (lines 229-236).
The sensory subchapter needs to be expanded to include the selection steps and performance of the reviewers. A good sensory panel should provide accurate, discriminate and accurate results. Ideal group performance is achieved when each group member distinguishes between products (high product variabilty), achieves the same values several times ("low within sessions"). However, he agrees with the other panel members on the sensory characteristics within a given tolerance (small difference between panelist’). By monitoring and monitoring performance, panel members and panels can be distinguished, their results are permanent, repeatable. Performance can be measured with performance indicators. The quality of the sensory data is determined by a qualified and expert sensory panel, and its members, and their performance must be constantly monitored. This is best done with software support, for example: PanelCheck V1.4.0., Sensominer (R-project), etc.
-The sensory panel used for the present study, is a Trained Sensory Panel of the University of Évora, which has a long-term experience in evaluating these kind of dry-cured meat products, as can be seen by the numerous publications of the group using this same sensory panel [1-4].
How did the Adhesiveness property get negative (Table 2)?
-Texture Profile Analysis (TPA) is a compression test. Adhesiveness is a traction force. Therefore, it appears as a negative value, because a traction force (Area 3 in the figure) is measured in a compression test. For more information, one of the papers cited in the manuscript can be consulted [5].
I found some inaccurate phrases or fonts that I suggest to improve:
In tables, values for a single parameter should be given to the same number of decimal places everywhere.
-Thank you for the correction. The same number of decimal places has been given for each parameter.
2.L.47.: World Health Organization instead of World Health Organisation
2.L.49.: World Health Organization instead of World Health Organisation
-The full name of the WHO has been amended. Thank you for the remark.
2.L.71.: 3.50% instead of (3,5%)
-The requested correction has been made
Reviewer #2:
Line 138: Due to the importance of sensory analyses in this study, It would be necessary not just quote the reference followed for the sample preparation but describe a little how was it performed. Also describe the panel and if they were trained. Why the authors did not perform preference tests?
-The methodology regarding sensory evaluations has been described in more detail in the revised version of the manuscript. Specifically, the referred International Standards (ISO) have been added (lines 229-236). Furthermore, the sensory panel used for the present study, is a Trained Sensory Panel of the University of Évora, which has a long-term experience in evaluating these kind of dry-cured meat products, as can be seen by the numerous publications of the group using this same sensory panel [1-4]. Preference tests are usually made with commercial purposes, and do not give much information. In the applied Quantitative Descriptive Analysis (QDA) Test, the aim was to evaluate the effect of each factor, genotype and salt content, on each sensory attribute. This kind of sensory analysis is generally more informative. With the purpose of identifying the samples preferred by assessors, we usually consider the “overall appreciation” attribute. In the present study, no significant differences were observed for this specific sensory attribute.
Line 222: Could you give an explanation for this?
-The positive correlation between salt content and aroma intensity has been reported before [6]. As mentioned in the revised version of the manuscript (lines 427-430), this could be due to the fact that salt reduction affects the generation of aroma active compounds. Specifically, Corral et al. [6] have reported a reduction in sulfur and acids, with a concomitant increase of aldehyde compounds, as a result of a reduction in salt content.
Line 233: How can authors affirm that acceptability was not significantly affected if they have not performed acceptability test? Overall appreciation is not a proper attribute to base half of the conclusion on that… Please, reformulate the conclusion and remove this part.
-You are absolutely right. The Conclusion has been reformulated accordingly.
References
- Fraqueza, M.J.; Laranjo, M.; Alves, S.; Fernandes, M.H.; Agulheiro-Santos, A.C.; Fernandes, M.J.; Potes, M.E.; Elias, M. Dry-Cured Meat Products According to the Smoking Regime: Process Optimization to Control Polycyclic Aromatic Hydrocarbons. Foods 2020, 9, 91, doi:10.3390/foods9010091.
- Laranjo, M.; Agulheiro-Santos, A.C.; Potes, M.E.; Cabrita, M.J.; Garcia, R.; Fraqueza, M.J.; Elias, M. Effects of genotype, salt content and calibre on quality of traditional dry-fermented sausages. Food Control 2015, 56, 119-127, doi:10.1016/j.foodcont.2015.03.018.
- Laranjo, M.; Gomes, A.; Agulheiro-Santos, A.C.; Potes, M.E.; Cabrita, M.J.; Garcia, R.; Rocha, J.M.; Roseiro, L.C.; Fernandes, M.J.; Fernandes, M.H., et al. Characterisation of “Catalão” and “Salsichão” Portuguese traditional sausages with salt reduction. Meat Science 2016, 116, 34-42, doi:10.1016/j.meatsci.2016.01.015.
- Laranjo, M.; Gomes, A.; Agulheiro-Santos, A.C.; Potes, M.E.; Cabrita, M.J.; Garcia, R.; Rocha, J.M.; Roseiro, L.C.; Fernandes, M.J.; Fraqueza, M.J., et al. Impact of salt reduction on biogenic amines, fatty acids, microbiota, texture and sensory profile in traditional blood dry-cured sausages. Food Chemistry 2017, 218, 129-136, doi:10.1016/j.foodchem.2016.09.056.
- Caine, W.R.; Aalhus, J.L.; Best, D.R.; Dugan, M.E.R.; Jeremiah, L.E. Relationship of texture profile analysis and Warner-Bratzler shear force with sensory characteristics of beef rib steaks. Meat Science 2003, 64, 333-339, doi:https://doi.org/10.1016/S0309-1740(02)00110-9.
- Corral, S.; Salvador, A.; Flores, M. Salt reduction in slow fermented sausages affects the generation of aroma active compounds. Meat Science 2013, 93, 776-785.
With best regards,
Maria João Fraqueza
FMV, ULisboa-CIISA
Portugal

Reviewer 2 Report
The general goals of this work was to study the effects on microbial growth and sensory properties of the reduction of salt in sausages. Overall the manuscript is well written, clear, and easy to understand. However, I have some concerns:
Line 138: Due to the importance of sensory analyses in this study, It would be necessary not just quote the reference followed for the sample preparation but describe a little how was it performed. Also describe the panel and if they were trained. Why the authors did not perform preference tests?
Line 222: Could you give an explanation for this?
Line 233: How can authors affirm that acceptability was not significantly affected if they have not performed acceptability test? Overall appreciation is not a proper attribute to base half of the conclusion on that… Please, reformulate the conclusion and remove this part.
Author Response

(The authors gave the same response as above.)

Round 2
Reviewer 1 Report
I suggest to accept this article in present form.